# Higher entropy observed in SARS-CoV-2 genomes from the first COVID-19 wave in Pakistan

**Najia Karim Ghanchi**[1◦]**, Asghar Nasir**[1◦]**, Kiran Iqbal Masood**[1]**, Syed Hani Abidi**[2]**, Syed Faisal Mahmood**[3]**, Akbar Kanji**[1]**, Safina Razzak**[1]**, Waqasuddin Khan**◉[4]**, Saba Shahid**[1]**, Maliha Yameen**[1]**, Ali Raza**[1]**, Javaria Ashraf**[1]**, Zeeshan Ansar**[1]**, Mohammad Buksh Dharejo**[5]**, Nazneen Islam**[1]**, Zahra Hasan**◉[1]***, Rumina Hasan**[1,6]*

1 Department of Pathology and Laboratory Medicine, The Aga Khan University (AKU), Karachi, Pakistan, 2 Department of Biological and Biomedical Sciences, AKU, Karachi, Pakistan, 3 Department of Medicine, AKU, Karachi, Pakistan, 4 Department of Pediatrics and Child Health, AKU, Karachi, Pakistan, 5 Department of Health, Government of Sindh, Karachi, Pakistan, 6 Faculty of Infectious and Tropical Disease, London School of Hygiene and Tropical Medicine, London, United Kingdom

◦ These authors contributed equally to this work.
* zahra.hasan@aku.edu (ZH); rumina.hasan@aku.edu (RH)

⬔ OPEN ACCESS

**Data Availability Statement:** Sequence data has been submitted to GISAID and accession numbers of the genomes are provided in S1 Table.

## Abstract

### Background

We investigated the genome diversity of SARS-CoV-2 associated with the early COVID-19 period to investigate evolution of the virus in Pakistan.

### Materials and methods

We studied ninety SARS-CoV-2 strains isolated between March and October 2020. Whole genome sequences from our laboratory and available genomes were used to investigate phylogeny, genetic variation and mutation rates of SARS-CoV-2 strains in Pakistan. Site specific entropy analysis compared mutation rates between strains isolated before and after June 2020.

### Results

In March, strains belonging to L, S, V and GH clades were observed but by October, only L and GH strains were present. The highest diversity of clades was present in Sindh and Islamabad Capital Territory and the least in Punjab province. Initial introductions of SARS-CoV-2 GH (B.1.255, B.1) and S (A) clades were associated with overseas travelers. Additionally, GH (B.1.255, B.1, B.1.160, B.1.36), L (B, B.6, B.4), V (B.4) and S (A) clades were transmitted locally. SARS-CoV-2 genomes clustered with global strains except for ten which matched Pakistani isolates. RNA substitution rates were estimated at $5.86 \times 10^{-4}$. The most frequent mutations were 5' UTR 241C > T, Spike glycoprotein D614G, RNA dependent RNA polymerase (RdRp) P4715L and Orf3a Q57H. Strains up until June 2020 exhibited an overall higher mean and site-specific entropy as compared with sequences after June. Relative entropy was higher across GH as compared with GR and L clades. More sites were under selection pressure in GH strains but this was not significant for any particular site.

**Funding:** Funding support from Health Security Partners, USA was received by RH. Funding support through a University Research Council Grant, Aga Khan University was received by ZH. The funders had no role in study design, data collection and analysis, decision to publish or preparation of the manuscript.

**Competing interests:** The authors have declared that no competing interests exist.

## Conclusions

The higher entropy and diversity observed in early pandemic as compared with later strains suggests increasing stability of the genomes in subsequent COVID-19 waves. This would likely lead to the selection of site-specific changes that are advantageous to the virus, as has been currently observed through the pandemic.

## Introduction

Severe Acute Respiratory Syndrome coronavirus 2 (SARS-CoV-2), the causative agent of COVID-19 was first reported in Wuhan, China [1] and has been to date reported in 190 million cases globally. SARS-CoV-2 RNA belongs to the *Sarbecovirus* subgenus of *Betacoronavirus* [2].

The first Wuhan strain identified in January 2020 was an L clade isolate and by the middle of 2020, SARS-CoV-2 was seen to diversify into S, V and G and its sub clades [3, 4]. SARS-CoV-2 has 8 coding and 6 non-coding genes and earlier, the greatest genetic variation observed in Nucleocapsid (N) and Orf1ab regions [5] however, since the end of 2020, evolution in Spike glycoprotein (S) regions has resulted in a number of variants of concern which are spreading rapidly across the globe [6]. Variants observed in Orf1ab, Orf3a, N and S genes are associated with evolutionary changes and thirteen signature single nucleotide variations (SNVs) divide strains into L, S, V, I, and G and its subclades as defined by GISAID [7]. Additional mutations have further characterize SARS-CoV2 genomes into GH, GR and GV clades [8].

In Pakistan, the first wave of COVID-19 occurred between March and July 2021, with a peak in mid-June where 4000–6000 positive COVID-19 cases were diagnosed each day [9]. There was a decrease in cases with a low in early September followed by a resurgence of COVID-19 cases with a second wave from October 2020 until January 2021. There was a brief reprieve in February with a third wave of cases between March and May 2021 [10]. COVID-19 cases started to rise again in June and currently in July 2021, Pakistan is well into the fourth wave [11]. Up to 17 July 2021, approximately 986,668 COVID-19 cases have been diagnosed with 22,760 deaths [12]. Of these, 354,103 (36%) COVID-19 cases were from Sindh province with 21% were from the city of Karachi. Provincial distribution of cases was found to be 349,890 (35%) from Punjab, 140,293 (14%) from Khyber Pakhtunkhwa Province (KPK), 84,399 (9%) from Islamabad Capital Territory (ICT), 28,884 (3%) from Baluchistan, 21,811 (2%) from Azad Jammu Kashmir and 7288 (1%) from Gilgit Baltistan. To date, the case fatality rate (CFR) for SARS-CoV-2 in Pakistan has been 2% with some regional variations [13]. Morbidity due to COVID-19 has been relatively lower as compared with many other countries [14, 15]. This could be due to both pathogen and host-related factors.

Study of genomic variation of SARS-CoV-2 strains helps understand the epidemiology of COVID-19. Global SARS-CoV-2 genomic sequencing efforts have contributed data of thousands (2,503,415) of strains into public database such as, GISAID and Nextstrain, NCBI SARS-CoV-2 Resources (https://www.ncbi.nlm.nih.gov/sars-cov-2/). These have allowed the interrogation of viral diversity with associated disease transmission in different countries [16].

There is limited genomic epidemiological data available for SARS-CoV-2 strains in Pakistan. The first introduction of SARS-CoV-2 was made by a traveller from Iran in March 2020. The pandemic was initially associated with travelers but local transmission was identified within two weeks of the first known COVID-19. G and S clade strains were shown to be

present in the first wave of COVID-19 in Pakistan [17]. Genetic studies of SARS-CoV-2 from the second wave in Pakistan have identified B.1 and its sub-clades together with B.6 lineage strains to be present [18].

Our clinical laboratory at the Aga Khan University Hospital, Karachi, Pakistan tested 196,588 respiratory samples for SARS-CoV-2 by PCR and reported 40,709 (21%) PCR positive COVID-19 cases between March and December 2020. Here we investigated the genomic diversity of SARS-CoV-2 studying the phylogeny of isolates from the first COVID-19 wave and compared them with later strains. Further, we studied diversity, site-specific mutation and entropy across the genome in strains isolated before and after June 2020, representing earlier and later periods.

## Materials and methods

### Sample selection

This study was approved by the Ethical Review Committee at the Aga Khan University (AKU), Karachi, Pakistan as a waiver of informed consent for the study. Samples used were those archived as de-identified samples previously reported positive for SARS-CoV-2 by the Clinical Laboratories, Aga Khan University Hospital (AKUH), Karachi, Pakistan. Laboratory data (including age, gender) was utilized where available.

Nasopharyngeal swab specimens were confirmed positive for SARS-CoV-2 by reverse transcription (RT) polymerase chain reaction (PCR) using the SARS-CoV-2 Cobas 6800 Roche assay at the Section of Molecular Pathology, AKUH, Karachi, Pakistan. A random, convenience sampling was done to include samples from March until October 2020.

Inclusion criteria were; samples amplified at CT value of 30 and below were selected, specimens collected between March and October 2020. Exclusion criteria were: samples with a crossing threshold (CT) value greater than 30.

### Sequencing of SARS-CoV-2 strains

A total of 70 samples were selected for SARS-CoV-2 genome sequencing. RNA was extracted from selected specimens using the QiaAmp RNA minikit (Qiagen, USA) and used as input for NGS library preparation.

In total, we successfully sequenced thirty-two SARS-CoV-2 specimens (twenty-one whole genome sequences and eleven partial genome sequences). Whole genomes of eight isolates were available through sequencing using the Nextera XT DNA Library Preparation kit (Illumina) was used for. As described previously, first-strand cDNA was synthesized with Super-Script III Reverse Transcriptase (SSIII), Thermo Fisher Scientific, USA, followed by Second strand cDNA synthesized with DNA polymerase I, Large Fragment, Klenow (Invitrogen, USA), (11). We obtained sequence data for twenty-four SARS-CoV-2 isolates using the Tru-Seq® Stranded Total RNA Library Preparation kit (Illumina). All normalized libraries were pooled and spiked with PhiX control prior to sequencing was performed on the Illumina Miniseq platform using a 300 cycle Miniseq Reagent Kit v2 (Illumina). In total we obtained twenty-one full and eleven partial genome sequences.

NCBI/GenBank accession numbers for twenty-one whole genomes deposited at https://www.ncbi.nlm.nih.gov/ are: MT730114, MT730115, MT731278, MT730116, MT730117, MT729995, MT731277, MW426405, MW490572, MW428254, MW433685, MW433687, MW433690, MW433716, MW433718, MW43372, MW433723, MW433725, MW433736, MW433740 and MW433741 (S1 Table).

## Variant calling and phylogenetic analysis

We performed whole genome analysis on thirty-two sequences from this study in addition to fifty-eight SARS-CoV-2 genomes from Pakistan available from GenBank (S1 Table). These comprised 79 full-length and 11 partial genomes. Fifty-four full length genomes were those isolated in the period March until June, 2020. Twenty-five full length genomes were from between July and October, 2020.

FASTQ files were aligned to the SARS-CoV-2 virus reference genome Wuhan-1 (NC_045512.2) by BWA [19]. SAM and BAM files were sorted using Samtools and variants were called using BCF tool-mpileup v-1.10.2. Additional variants were identified and annotated using Variation Identification online tool from China National Bioinformatics Center Novel Coronavirus Resource (2019nCoVR) (https://bigd.big.ac.cn/ncov/online/tool/variation) to enhance the variant call confidence. Further, variants were annotated for effect on protein-coding region by the mutation using the customized build database of nCoV-2 using SnpEff v-5.0c. [20]. The effect on protein-coding by the mutation is determined by SIFT impact scores i. e Low (0.05–1.0 –Tolerated benign), Medium (0.00–0.05 –considered to be deleterious), High (0.0 –highly deleterious) (https://faculty.washington.edu/wjs18/GS561/cSNPs_lab.html).

Phylogenetic analysis was performed using the 79 available full length SARS-CoV-2 genomes (S1 Table) along with the 449 full-length SARS-CoV-2 reference sequences (S1 File) from different pandemic countries obtained from the NCBI SARS-CoV-2 Resources (https://www.ncbi.nlm.nih.gov/sars-cov-2/) were subjected to Multiple Sequence Alignment (MSA) along using MAFTT online server [21]. The MSA was subsequently used to generate a Maximum Likelihood (ML) phylogenetic tree using PhyML 3.0 (http://www.atgc-montpellier.fr/phyml/) with a GTR-based nucleotide substitution model and aLRT SH-Like branch support. The root of the tree and branch length variance was determined using the TreeRate tool [22] by applying a generalized midpoint rooting strategy. The tree was visualized and edited in Figtree software (http://tree.bio.ed.ac.uk/software/figtree/). Mean and individual pairwise distance between SARS-CoV-2 sequences from our study and previously deposited Pakistani SARS-CoV-2 sequences was calculated using MEGA 7 [23].

For genomic epidemiology of Asian- strains focused sub-lineage group analysis of SAR-CoV-2 as of 18th January 2021, we downloaded 6,602 global complete sequences of SARS-CoV-2 along with the required metadata from the GISAID (https://platform.gisaid.org/epi3/) considering the following parameters: 1) genome length > 29,000 bps, 2) further assigns labels of high-coverage <1% Ns–undefined bases, and 3) A and B lineages and its sub-lineages identified by ncov-19 Pangolin Lineage identification tool (https://pangolin.cog-uk.io/).

The fasta files were used for phylogenetic tree reconstruction using NEXTSTRAIN's (https://www.nextstrain.org/) augur (https://www.docs.nextstrain.org/projects/augur/en/stable/) pipeline. Out of 6,602 whole SARS-CoV2 genome sequences, 2,101 qualified for the phylodynamic map. These included Asian (n = 825), European (n = 818), South American (n = 90), Oceanian (n = 16), North American (n = 156), and African (n = 196) sequences. Ancestral state reconstruction and branch length timing were performed with IQTree (http://www.iqtree.org/) and TreeTime [24]. Finally, the collection of all annotated nodes and metadata was exported to the interactive phylodynamic visualizing tool Auspice's (https://auspice.us/) in JSON format.

## Entropy and site selection (dN-dS) analysis

Genome-wide Shannon entropy and site selection (dN-dS) analyses were performed to evaluate genomic variability between SARS-CoV-2 genomes from Pakistan isolated up until June 2020 (representing strains from the first COVID-19 wave) and those isolated later. Further, we

also performed dN-dS entropy analysis for each clade of strains separately. However, as both entropy and site selection analysis require at least three taxa to give a meaningful output, these analyses could be performed for L, GH, and GR clade strains only and not the V and S clade strains in our study. Shannon entropy for each clade was carried out using Bioedit entropy function. The statistical significance in mean entropy value between 1st and 2nd wave sequences was evaluated using the paired T-test, while statistical significance in mean entropy value between different clades was evaluated using the One-way Anova test. Both tests were performed using 95% confidence interval and p<0.05 as significant value. The calculations were performed using GraphPad Prism tool.

For site selection analysis, codon alignment was performed using MEGA 7, using Muscle algorithm, and the codon aligned file was subsequently used for site selection using SNAP tool available at the Los Alamos Database (www.hiv.lanl.gov) and SLAC tool available at DataMonkey [25]. It is important to note that SLAC was pre-cited as the best model for our data set based on algorithm selection criteria available on DataMonkey website. Normalized dN-dS data was plotted and analyzed for the presence of significant positively or negatively selected sites.

# Results

## Description of SARS-CoV-2 isolates

We determined the genomic epidemiology of ninety SARS-CoV-2 strains isolated between March and October 2020. Overall, the strains were found to be belong to clades L (n = 13, 14.5%), S (n = 5, 5.5%), V (n = 2, 2.2%), O (n = 1, 1.1%), G (n = 1, 1.1%) clade with sub-types GH (n = 59, 65%) and GR (n = 9, 10%), S1 Table.

A month-wise distribution of the strains showed eleven isolates sequences from March, nineteen from April until May, forty-two from June until July and eighteen between August and October 2020. G and L clade strains were present throughout the period studied (S1 Fig). Strains belonging to the GH clade became predominant from April onwards until October. The frequencies of L and S clades reduced over the study period. GR and V clades were found in March and then in June- July. Notably, the diversity of clades was reduced over the study period in that by October only L, GR and GH (as the predominant clade) were present.

Four SARS-CoV-2 GH clades strains identified in March were from travelers from Iran and Turkey. Apart from these, the remaining eight-six strains were from across the four different provinces of the country without any known travel history. These cases of likely local transmission were from the province of Sindh (n = 64: clades G, n = 1; GH, n = 46; GR, n = 1, L n = 10; S n = 3; V n = 2), Punjab (n = 5; clade GR, n = 5), ICT (n = 17; S n = 2; GH n = 12; GR n = 3) and KPK (n = 4, GH n = 1; L n = 3).

## SARS-CoV-2 lineage analysis

**Phylogenetic analysis.** Phylogenetic analysis of 79 full length SARS-CoV-2 genomes from Pakistan and 449 global isolates was conducted (Fig 1). Of the 21 genomes sequenced in this study (AKU), seven clustered with sequences from Saudi Arabia and India (Fig 1, orange and purple branches, respectively). Four clustered with sequences from US (Fig 1, pink branches), and ten (AKU-2, -3, -21, -24, -25, -26, -33, -46, -47, -56) clustered with previously deposited sequence from Pakistan (Fig 1, green color branches). These ten AKU strains were all from Karachi (S1 Table), suggesting that the viral strains circulating in the city were predominantly similar to those circulating in other parts of Pakistan. The mean pairwise genetic distance between our sequences; sequences previously deposited from Pakistan, and sequences from

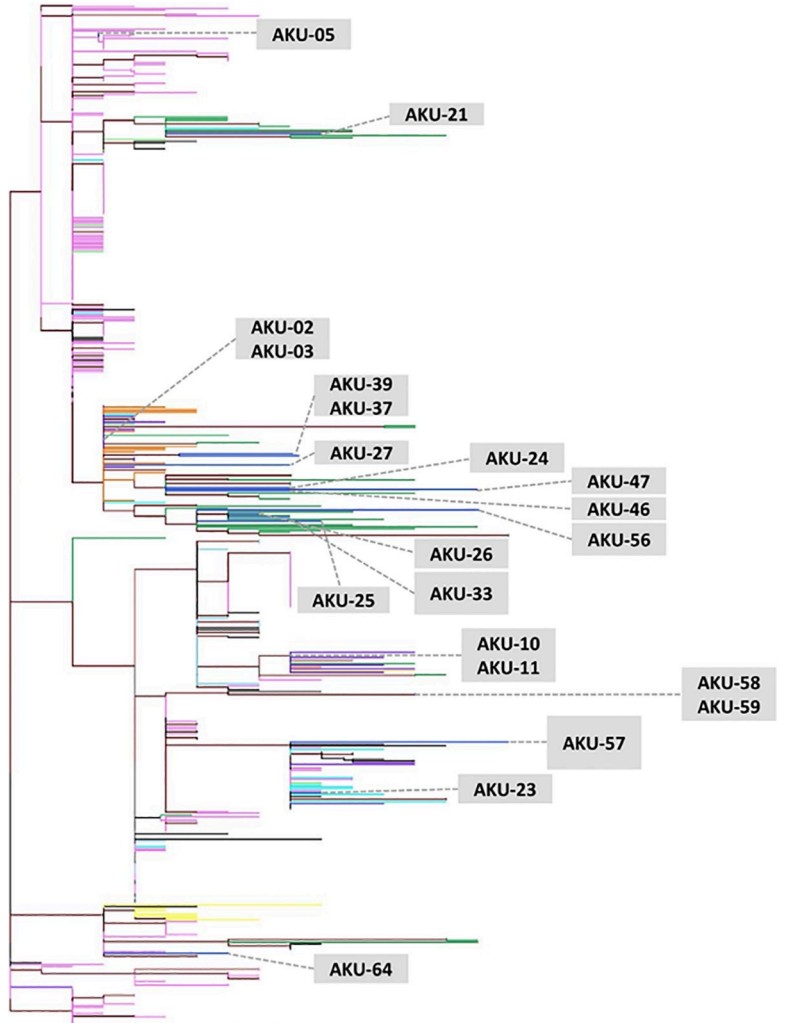

**Fig 1. Maximum-Likelihood phylogenetic tree of SARS-CoV-2 sequences from Karachi.** The tree was constructed using 21 genomes from this study (AKU) (S2 Table) along with 58 other Pakistani and 449 full-length SARS-CoV-2 reference sequences (S1 File). AKU study sequences are indicated in blue, while other Pakistan sequences are shown in green. AKU sequences clustered with those from India (purple), Saudi Arabia (orange), United States (pink), China (light blue), Australia (turquoise), Bangladesh (yellow), France (light green) and also with other sequences from Pakistan (green). The root of the tree was determined using TreeRate tool by applying generalized midpoint rooting strategy. Nodes with significant (>0.90) aLRT-SH like support values are colored maroon. The tree was visualized and edited in Figtree software.

India, Saudi Arabia, USA, and Australia was found to be 0.00, indicating phylogenetic relatedness between the genomes.

**Nextstrain analysis.** Expanded phylogenetic analysis to examine the genetic divergence of strains was conducted against a representative subset of 825 Asian SARS-CoV-2 genomes present in Nextstrain database using data for the period January 2020 to January 2021. Pakistani SARS-CoV-2 genomes aligned throughout the phylogenetic tree indicating multiple introductions of the SARS-CoV-2 in the country (Fig 2). 71 sequences from Pakistan are displayed in the global phylogenetic tree of SARS-CoV-2, primarily classified on the basis of clade (19A, 19B, 20A, 20B, 20C and 20D). Out of 71 sequences from Pakistani population, 48 sequences were clade 20A, seven were clade 20B, one was clade 20C and seven belonged to clade 20D. Further, there are 10 and 3 Pakistani sequences, of clades 19A and 19B respectively (S1 Table).

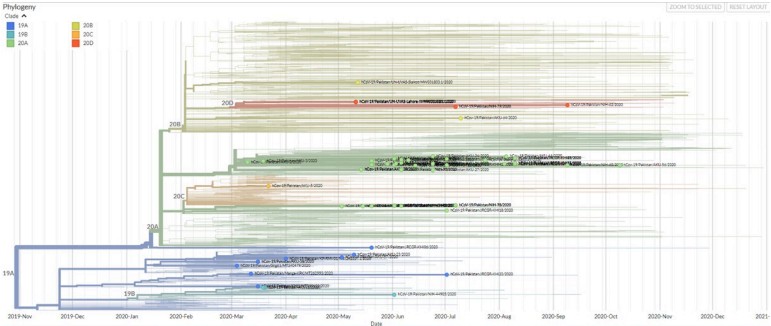

**Fig 2. Time-resolved phylogenetic distribution of genomic epidemiology of SAR-CoV-2 focused on Asian subsampling (Screenshot of the current Nextstrain display in SVG format are protected by CC-BY license).** Tree option layout is selected as rectangular, branch length is set as time interval while branch labels on the basis of phylogenetic clades (19A, 19B, 20A, 20B, 20C and 20D) showing SARS-CoV-2 genomes from Pakistan highlighted in green.

Pangolin (Phylogenetic Assignment of Named Global Outbreak Lineages) classification identified a majority of B and B sub-lineages isolates (93.6%) with five ancestral lineage A strains. The five sequences from March with a travel history to Iran and Turkey belonged to A, B.1 and B.1.255 lineages. The most commonly observed SARS-CoV-2 lineages were B.1 (n = 16, 20%), B.1.1.1 (n = 7, 8%), B1.160.4 (n = 4, 5%), B.1.255 (n = 3, 3.7%), B.1. 36 (n = 13, 16%), B.1.471 (n = 14, 17.7%) and B.6 (n = 4, 5%). Additionally, B and its sub-lineages B.1.1, B.1.1.105, B.1.260, B.1.275 and B.4 were also identified.

Subsequently, Nextstrain analysis was run for global temporal and spatial classification of the isolates. A, B, C and D lineages were identified comprising 19A; L and V clades (n = 10), 20A or, GH clade (n = 20), 19B or, S clade (n = 5), 20B or, GR clade (n = 6) and 20C or, GH clade (n = 35) and 20D or, GR clade (n = 3) Fig 2, S1 Table. Of these, 20A (28%), 20C (44%) and 19A (13%) were predominant clades. In March, the strains introduced by travelers from Turkey and Iran were 19B, 20A and 20C lineage isolates. We observed types 19B to exhibit coincident time lineages with those from Saudi Arabia. Types 20A along with 20B were found to persist up until October 2020.

During the first wave of COVID-19 the dominant types were 20A and 20C across all locations (S2 Fig). The diversity of SARS-CoV-2 clades was most apparent in Sindh (19A, 19B, 20A, 20B, 20C) and ICT (19B, 20A, 20C, 20D). Whilst, all strains Punjab belonged to 20B. Those from KPK were either 19A or 20A.

### Estimation of mutation rates

We investigated the stability of the SARS-CoV-2 genomes by determining their mutation rates. Pairwise time tree distances and substitution rates were estimated 5.68 x $10^{-4}$ substitution per site per year (16.98 substitution per year) for the Pakistan tree compared to $8 \times 10^{-4}$ substitution per site per year (23.92 substitution per year) globally, (Fig 3). This demonstrates the substantial variation in SARS-CoV-2 phylogenies between March and October, 2020.

### Variant analysis of SARS-CoV-2 strains

Variant analysis revealed 257 SNVs comprising of 18 non-coding (2 in 5'UTR and 16 in 3'UTR), 138 non-synonymous and 101 synonymous variants (S2 Table). Compared with the Wuhan-Hu-1 reference (clade L) the average variation observed was 11 SNPs ranging between 2–19 variants per genome in this study. The most frequent variants were observed at positions

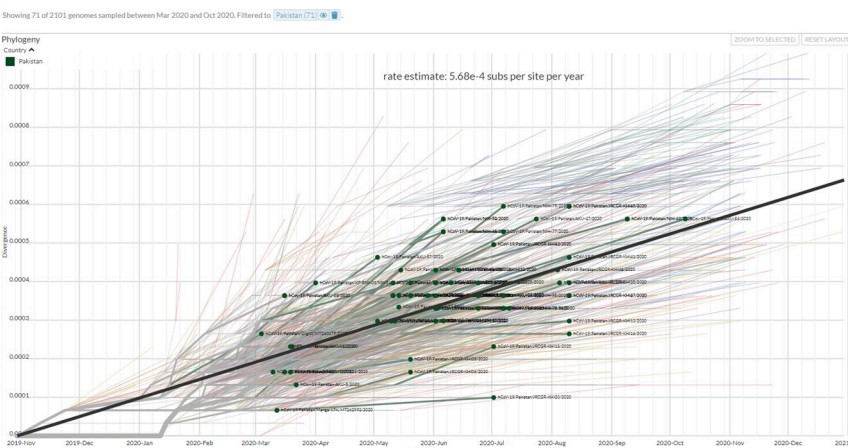

**Fig 3. Estimation of divergence in Pakistani SARS-CoV-2 strains over time.** Isolates from Pakistan were matched with a sub-sample from Asian over time. The graph shows an estimate of divergence within genomes over the time period.

5' UTR; 241C>T (78.8%), S gene D614G (76%) Orf1ab; Nsp3 924F (74%), Orf1ab; RdRp P4715L (72%), Orf3a Q57H (70%), exonuclease 1ab 6205L (51%), M gene 71Y (51%). N gene R209I (20%), Orf1ab; nsp6 L3606F (15%), N geneS194L (15%), Orf1ab; Nsp3 Q2702H (13%), Fig 4.

Of the mutations in Orf1ab, six were associated with evolutionary changes; 8782 (nsp4 2839S), 14408 (RdRp P4715L), 1397 (nsp2 V378I), 3037 (nsp3 924F) and 1059 (nsp3 T265I). G clade strains displayed the greatest number of variations with lineage associated variants including nsp3 924F, RdRp P4715L and S D614G. A GH clade strain from May had L lineage associated mutation (Orf1ab L3606F and P323L) in addition to those typical of its clade.

Seventy-nine different non-synonymous SNVs were observed in the Orf1ab gene encoding the non-structural proteins with the greatest number observed in the nsp3 (n = 21), nsp2

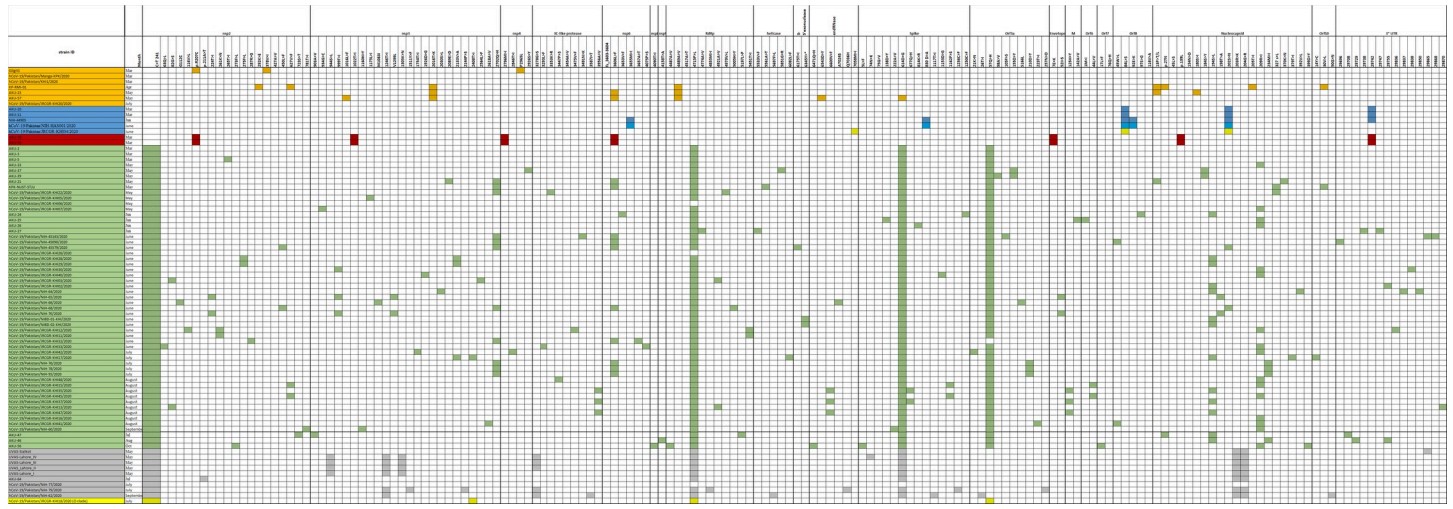

**Fig 4. Genetic variants found in SARS-CoV-2 genomes.** Clade and time-wise association of genome variants identified in seventy-nine full-length SARS-CoV-2 isolates are identified. Variations of upstream, downstream and non-synonymous SARS-CoV2 genome are presented in the grid format. The isolates represented clades L (n = 7), S (n = 5), V (n = 2), GH (n = 55), GR (n = 9) and O (n = 1). Colors represent clades as, L (orange yellow) S (blue), V (red), G (dark green), GH (light green), GR (grey) and O (yellow).

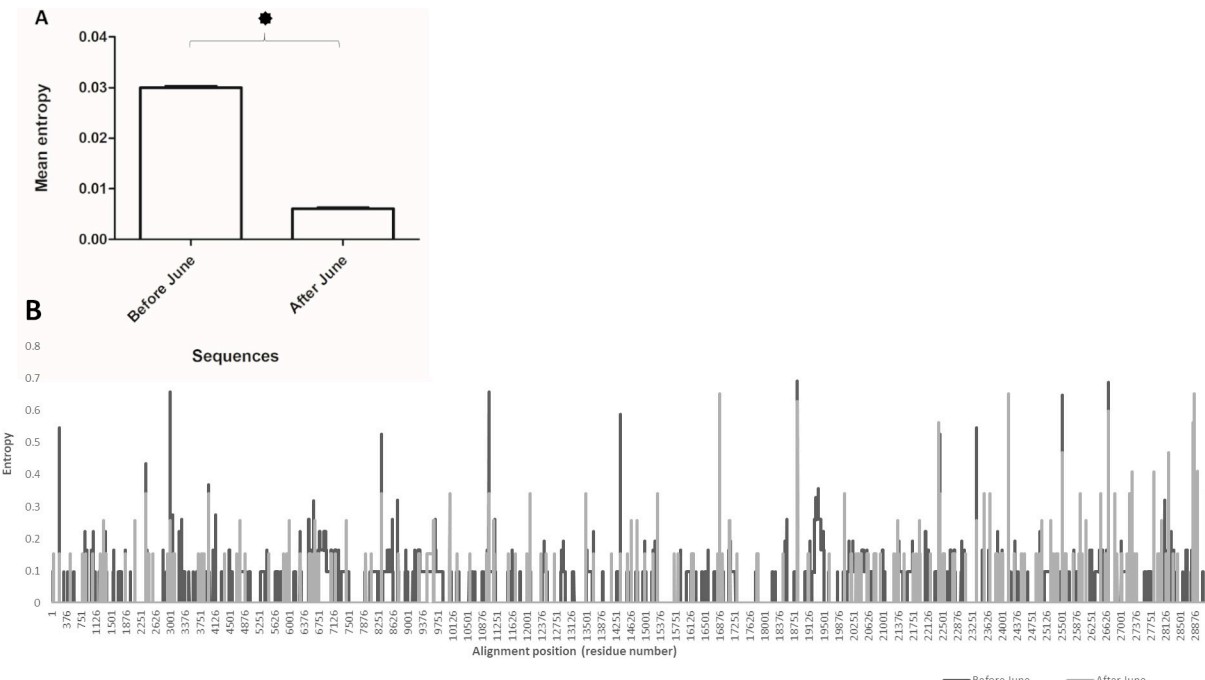

**Fig 5. Genome-wide site-by-site entropy analysis for SARS-CoV-2 sequences from before and after June, 2020.** The figure shows A) Mean entropy and B) site-by-site entropy for sequences collected before and after June 2020. '*' in 4A indicates statistically significant (p<0.001) difference between the means. Error bars shown standard error of mean.

(n = 16) and RdRp (n = 12) regions, respectively. Within structural proteins, 59 different non-synonymous SNVs were observed, with greatest numbers seen in N gene (n = 21), S gene (n = 15) and ORF3a (n = 7). In N gene, the most frequent ns-SNV were R209I and S194L where in the S-gene the most frequent ns-SNV was D614G (S2 Table).

## Entropy and site selection pressure analysis

In the next step, we investigated the correlation between genetic diversity, entropy and site selection pressures across SARS-CoV-2 genomes. Entropy gives an estimate of the probability of acquiring mutations in a given set of genomic sequences [26].

The first COVID-19 wave in Pakistan peaked in June 2020, hence, we compared SARS-CoV-2 strains within our study period as those isolated before and after June 2020. There were fifty-seven genomes from March until the thirtieth of June. Thirty-three genomes were isolated from between first of July and until the end of October.

We first investigated the difference in genome-wide entropy between Pakistani SARS-CoV-2 sequences from before and after June, representing samples from the first COVID-19 wave in Pakistan and later isolates. Our results revealed that sequences collected before June 2020 exhibited an overall higher mean (Fig 5A, p<0.001) and site-specific entropy (Fig 5B) across genome as compared with sequences collected after June, 2020. Indicating, greater overall stability in the genomic pressures of SARS-CoV-2 strains in the second wave. Analysis of entropy prevalent clades L, GR and GH revealed clade GH to have an overall higher mean entropy (p<0.001) across the genome as compared with GR and L clade genomes (Fig 6A). In the GH clade genomes we observed multiple sites exhibiting entropy above 0.5 and up to 0.9 (Fig 6B). This was followed by second highest entropy across genomes of the L clade (p<0.001), while

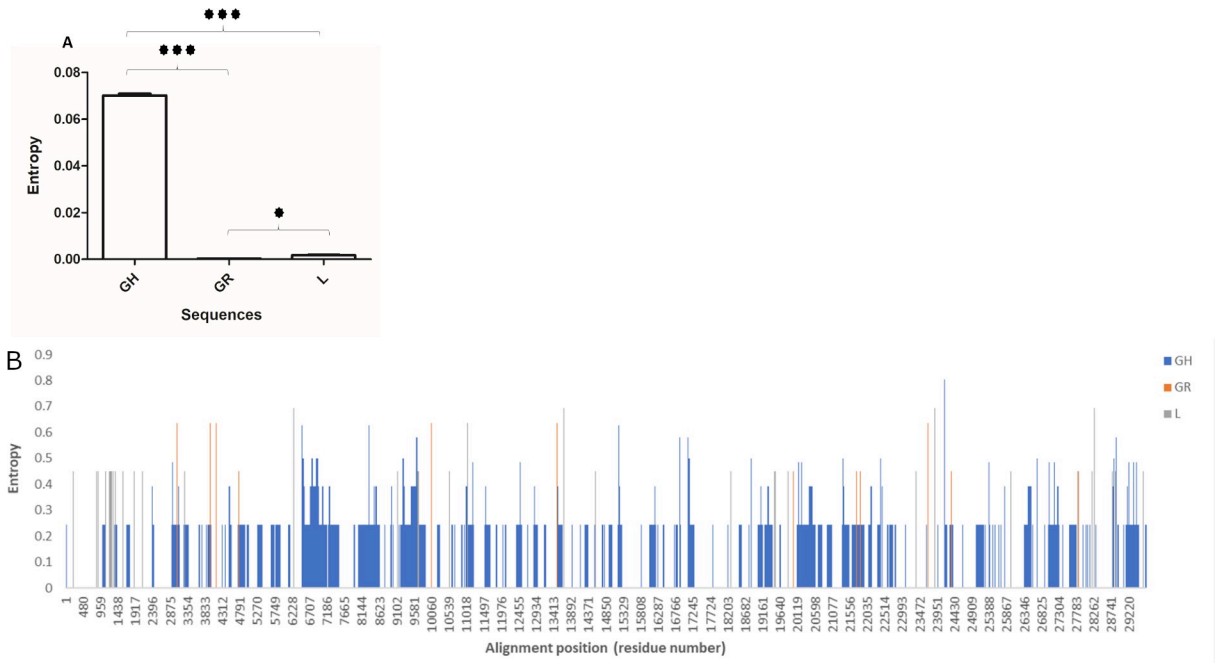

**Fig 6. Genome-wide site-by-site entropy analysis for prevalent clades L, GR and GH.** The figure shows A) Mean entropy and B) site-by-site entropy for GH, GR and L clade sequences A) The asterisk above the bars show statistically significant (*** = p<0.001; * = p<0.05) difference between the means. Error bars shown standard error of mean.

GR clade (p<0.05) genomes had fewer high-entropy sites and an overall lower pan-genomic entropy as compared to clade GH genomes.

A subsequent site-selection analysis revealed several genomic locations under positive or negative selection in sequences before and after June 2020 (Fig 7A and 7B), however, none of the sites were under statistically significant selection pressure. In agreement with the entropy analysis, clade GH exhibited higher number of sites under selection pressures, followed by clade L and GR (S3–S5 Figs). However, none of the sites in any clade were found to be under statistically significant selection pressure.

## Discussion

Our study provides insights into the SARS-CoV-2 strains circulating between the first wave of COVID-19 in Pakistan and those after this period. It associates genetic diversity of the mostly commonly found GH clade strains with greater relative entropy without a site-specific selection bias.

SARS-CoV-2 genomic surveillance data available at GISAID provide key information regarding the viral diversity of SARS-CoV-2 genomes across the globe. New mutations are continually being identified but the biological effects of most of them remain unclear. Recently reported spectrum of mutations mainly spike 69/70 deletion, E484K and N501Y variants in B.1 lineage strains, have revealed a role for mutations in driving virulence of the virus by impacting host infection, transmission, diagnostics, and vaccine escape [27, 28].

Coronavirus replication is error-prone, with a high mutation rate and a nucleotide mutation rate estimated at $4 \times 10^{-4}$ substations/site/year [29]. For SARS-CoV-2, the substitution rate is considerably higher i.e., $9.90 \times 10^{-4}$ substitutions/site/year ($6.29 \times 10^{-4}$ to $1.35 \times 10^{-3}$) based on one study, whilst it is thought to be $5.3504 \times 10^{-3}$ and $5.35 \times 10^{-3}$ based on other studies [30]. We found the SARS-CoV-2 genome substitution rate to be $5.68 \times 10^{-4}$, which lies within

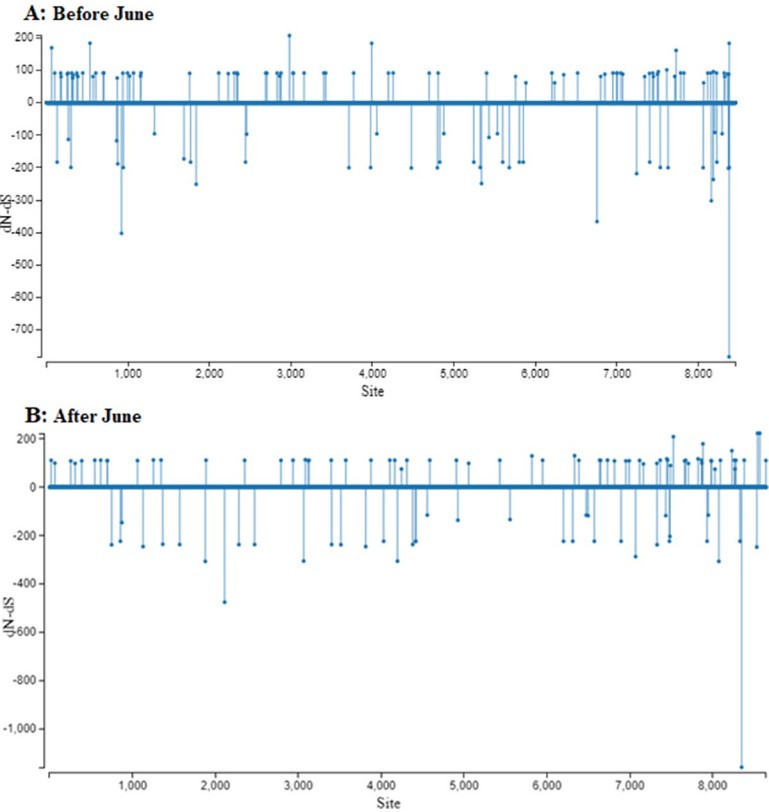

**Fig 7. Genome-wide site-by-site selection pressure analysis for SARS-CoV-2 sequences from before and after June, 2020.** Selection pressure (relative number of non-synonymous substitutions minus synonymous substitutions (dN–dS)) on each codon is shown for sequences A) before June 2020 and B) after June 2020.

the range previously calculated. The variability in substitution rates in different studies might be due to estimation model bias or rapid evolution of virus [30].

Pakistan SARS-CoV-2 genomes were genetically diverse and clustered with those from Saudi Arabia, India, USA, Australia, Italy and China. Comparison with a sub-sample of Asian isolates revealed that there was clustering with virus genomes from Japan, which might indicate that strains from these countries might be genetically similar and evolving at a similar rate. Ten sequences from this study clustered with previously deposited sequenced from Pakistan suggesting limited diversity between strains circulating in Pakistan. However, this data may also be impacted by the limited genomic information on SARS-CoV-2 strains currently available for the country.

We show that in Karachi there was a shift from S, L, V and G clades in March to predominantly the GH clade strains from May onwards [17]. Traveler- associated strains belonged to S and GH clades. GH (B.1.255) and S (A) clade strains were introduced through travelers from Iran and Turkey, corroborating with previous reports [31]. An L lineage isolate (B.6) was identified from a religious pilgrim who attended a super spreader event in Punjab Province which had visitors from abroad including China. L lineage isolates exhibiting L3606F mutations were first reported on January 2020 in China and associated with super spreader events in the USA, Singapore, Japan, and Europe [3]. GH (B.1., B.1.36, B.1.160, B.1.255, B.1.275) and L (B, B.6) strains associated with local transmission persisted between March and October. Data from Karachi concur with previous reports where B.1 sub-lineages have been reported [32].

Variant analysis revealed Orf1ab, nsp3 and N region to have the highest number of mutations, as shown previously [5, 33]. N gene P13L was present in both S clade and A lineage isolates in this study, it has been previously reported from the UK and Australia [34]. N gene variants at codon 202, 203, 204 were in 9% of the isolates whereas and 209 were found in 20% of isolates further reinforcing the increased mutability observed in this protein; these variants are associated with a split of G into GH and GR clades.

The most common non-synonymous variants observed were S D614G, Orf1ab RdRp gene P4715L and, Orf3a Q57H which have previously been reported from Europe and the United States [33]. Five isolates with the ORF8 L84S mutation also had N gene S202N. The L84S mutation is common amongst S clade isolates found in Europe where it was found to co-evolve with mutations such as P323L [34]. The L lineage initially split equally into G and V subclades, with G reaching 50% of the viruses isolated in March 2020 and then splitting further into GR and GH subclades [35]. We observed a GH isolate from May to have Orf1ab L3606F mutation co-occurring with P323L. L3606F is not typical of GH the clade, and is a transitory mutation defining L to S clade divergence, possibly identifying further evolutionary transitions in this isolate [3].

The D614G mutation in S gene was present in the majority (76%) of the genomes studied here and reported globally [17]. The Spike glycoprotein assists viral entry into host cells by binding to the ACE2 receptor [36, 37]. The substitution of Glycine at the 614 mutations may have introduced structural instability into the spike protein [38]. It has been associated with increased virulence and transmission of SARS-CoV2, most probably due to a higher viral load [39, 40]. G clade isolates comprise the predominant proportion of isolates in Europe and North America. The D614G mutation has also been associated with greater mortality observed in Belgium, Spain, Italy, France, Netherlands and Switzerland [41]. Importantly, eight different singly occurring S gene ns-SNV were observed in the study isolates. One strain had two S gene mutations, D614G and L5F. This double mutation has been shown to increase the infectivity of human cell lines as compared to the reference Wuhan strain [6].

The RdRp variant P4715L was present in 72% of all isolates. P4715L and D614G containing isolates have shown significant positive correlations with fatality rates in many countries [42]. RdRp is the target for polymerase inhibitors and mutations in RdRp can potentially decrease drug-RdRp complex binding affinity leading to resistance and a differential effect of antiviral treatments [43]. The frequency of P4715L has been shown to differ between SARS-CoV-2 genomes isolated from the USA (63.0%) and China (11.2%) and have been attributed to the differential efficacy observed in the clinical trials of the antiviral agent Remdesivir [44].

Variants in the non-coding 5'UTR and 3' UTR regions of the SARS CoV2 virus have been reported to affect viral replication and transcription. The 241C > T mutation has been reported to result in the strong binding of *TARDBP* (*RNA/DNA-binding protein*) to the 5'UTR region of the SARS CoV2 virus [45]. This variant has been implicated in facilitating the translation of viral proteins resulting in its effective propagation within the human host. Interestingly, variant (241C > T) of the 5'UTR region often coexists with spike glycoprotein variant (S protein, D614G) [46]. This coexistence is also evident in our study as we found 4 GH clade strains and a V clade strain to have the 5' UTR variant +241 C>T. Two S clade strains had the 3' UTR + 29742 G>A. This mutation is thought to affect binding of the miR-1307 which regulates the stability of the RNA at a post-transcriptional level and therefore, it potentially weakens the host immune response against the virus [47].

SARS-CoV-2 genomic surveillance data into GISAID provides key information regarding the viral diversity of genomes across the globe. New mutations are continually being identified but the biological effects of most of them remain unclear. Recently reported spectrum of mutations mainly spike 69/70 deletion, E484K and N501Y variants in B.1 lineage strains, have

revealed a role for mutations in driving virulence of the virus by impacting host infection, transmission, diagnostics and vaccine escape [27, 28].

Coronavirus replication is error-prone, with a high mutation rate estimated at $4\times10^{-4}$ substations/site/year [29]. For SARS-CoV-2, the substitution rate is considerably higher i.e., $9.90 \times 10^{-4}$ substitutions/site/year ($6.29 \times 10^{-4}$ to $1.35 \times 10^{-3}$) based on one study, whilst it is thought to be $5.3504 \times 10^{-3}$ and $5.35 \times 10^{-3}$ based on other studies [30]. We found the SARS-CoV-2 genome substitution rate to be $5.86 \times 10^{-4}$, which lies within the range previously calculated. Such a high substitution rate can have a significant impact on the evolution of viral clades and emergence of new SARS-CoV-2 variants. The variability in substitution rates in different studies might be due to estimation model bias or rapid evolution of virus [30].

We investigated the correlation between viral diversity and sequence variability using entropy and site selection analysis. Genomes from clade GH were found to have an overall higher pan-genomic entropy as well as higher evidence of pervasive selection as compared with clades GR and L. Increase site specific pressure and entropy, especially across Orf1ab and N regions in GH clade is consistent with the mutation data, and might suggest that the high selection pressure of GH clade is attributed to high transmission opportunities and/or dissemination into genetically diverse population observed in the transient phase of viral evolution before the virus adapts to the population/host selection pressures [48].

A limitation of this study is the relatively small sample size. However, as the Aga Khan University Hospital initiated SARS-CoV-2 PCR based diagnostics in Karachi, Sindh at the start of the pandemic in February 2020, a large proportion of test samples from Sindh province were referred to our laboratory. Therefore, the genomes described here are considered representative of the SARS-CoV-2 strains circulating in the province during and after the first COVID-19 wave.

## Conclusions

Our results highlight the value of genomic surveillance in understanding the evolution of SARS-CoV-2 strains in Pakistan. Our data shows that the predominant GH clade strains were more genetically diverse but did not display site specific pressures. Importantly, there was a higher mean and site-specific entropy in SARS-CoV-2 genomes isolated before as compared with after June 2020. This suggests that the viral genome achieved stability after the initial early period of the pandemic. It will be important to continue genomic epidemiological studies surveillance of SARS-CoV-2 to understand transmission patterns in the country.

## Supporting information

**S1 Fig. Temporal distribution of SARS-CoV-2 clades identified between March and October 2020.** L, S, V, O and G, GH and GR clades are depicted as a percentage of the total genomes analysed in March (n = 11), April—May (n = 19), June—July (n = 42), and August—October (n = 18). The y axis represent the percentage of each isolate of the total number of strains. The number of SARS-CoV-2 strains of each clade within each period is inset within the graph.
(TIF)

**S2 Fig. Provincial distribution of SARS-CoV-2 clades identified between March and October 2020.** Next strain 19A,19B,20A, 20B, 20C and 20D clades are depicted as a percentage of the total genomes analysed from Sindh (n = 11), Punjab (n = 11), Khyber Pakhtunkhwa Province (KPK), n = 11 and Islamabad Capital Territory (ICT), n = 11.
(TIF)

**S3 Fig. Genome-wide site-by-site selection pressure analysis for sequences from GH clade.** Selection pressure (relative number of non-synonymous substitutions minus synonymous substitutions (dN–dS)) on each codon is shown.
(TIF)

**S4 Fig. Genome-wide site-by-site selection pressure analysis for sequences from prevalent clades GR clade.** Selection pressure (relative number of non-synonymous substitutions minus synonymous substitutions (dN–dS)) on each codon is shown.
(TIF)

**S5 Fig. Genome-wide site-by-site selection pressure analysis for sequences from L clade.** Selection pressure (relative number of non-synonymous substitutions minus synonymous substitutions (dN–dS)) on each codon is shown.
(TIF)

**S1 Table. Description of SARS-CoV-2 cases from Pakistan.**
(XLSX)

**S2 Table. Description of variants found in 90 SARS-CoV-2 isolates from Pakistan.**
(DOCX)

**S1 File. Accession numbers of samples used for phylogenetic analysis in Fig 1.**
(XLSX)

## Acknowledgments

We thank Drs. Roger Hewson and Barry Atkinson, Public Health England, UK for their assistance in establishing SARS-CoV-2 diagnostics at AKU. Thanks to the European Virus Archive —Global (EVAg), a European Union infrastructure project for making available control material for the study. Thanks for technical support to the Aga Khan University Hospital (AKUH) Clinical Laboratory sections of Molecular Pathology and Microbiology.

## Author Contributions

**Conceptualization:** Zahra Hasan.

**Data curation:** Najia Karim Ghanchi, Syed Hani Abidi, Akbar Kanji, Safina Razzak, Waqasuddin Khan.

**Formal analysis:** Asghar Nasir, Kiran Iqbal Masood, Syed Hani Abidi, Safina Razzak, Waqasuddin Khan, Javaria Ashraf.

**Funding acquisition:** Syed Faisal Mahmood, Zahra Hasan, Rumina Hasan.

**Investigation:** Najia Karim Ghanchi, Asghar Nasir, Kiran Iqbal Masood, Syed Faisal Mahmood, Akbar Kanji, Saba Shahid, Maliha Yameen, Ali Raza, Zeeshan Ansar, Mohammad Buksh Dharejo, Nazneen Islam, Zahra Hasan.

**Methodology:** Najia Karim Ghanchi, Asghar Nasir, Akbar Kanji, Waqasuddin Khan, Saba Shahid, Maliha Yameen, Ali Raza, Javaria Ashraf.

**Project administration:** Zahra Hasan.

**Resources:** Rumina Hasan.

**Supervision:** Zahra Hasan.

**Validation:** Akbar Kanji.

**Writing – original draft:** Najia Karim Ghanchi, Asghar Nasir, Syed Hani Abidi, Zahra Hasan.

**Writing – review & editing:** Syed Faisal Mahmood, Ali Raza, Zahra Hasan, Rumina Hasan.

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
