## [Decision Letter · Decision Letter 0]

9 Jun 2021

PONE-D-21-09819

Increased diversity with absence of site-specific selection bias in SAR-CoV-2 genomes from the early pandemic period in Pakistan

PLOS ONE

Dear Dr. Zahra,

Thank you for submitting your manuscript to PLOS ONE. After careful consideration, we feel that it has merit but does not fully meet PLOS ONE’s publication criteria as it currently stands. Therefore, we invite you to submit a revised version of the manuscript that addresses the points raised during the review process.

We look forward to receiving your revised manuscript.

Kind regards,

Sagheer Atta, Ph.D

Academic Editor

PLOS ONE

Journal Requirements:

2.Thank you for including your ethics statement:  "This work received approval by the Ethical Review Committee, Aga Khan University.".   

Please provide additional details regarding participant consent. In the ethics statement in the Methods and online submission information, please ensure that you have specified (1) whether consent was informed and (2) what type you obtained (for instance, written or verbal, and if verbal, how it was documented and witnessed). If your study included minors, state whether you obtained consent from parents or guardians. If the need for consent was waived by the ethics committee, please include this information.

3.In your Data Availability statement, you have not specified where the minimal data set underlying the results described in your manuscript can be found. PLOS defines a study's minimal data set as the underlying data used to reach the conclusions drawn in the manuscript and any additional data required to replicate the reported study findings in their entirety. All PLOS journals require that the minimal data set be made fully available. For more information about our data policy, please see http://journals.plos.org/plosone/s/data-availability.

Reviewers' comments:

Reviewer's Responses to Questions

**Comments to the Author**

1. Is the manuscript technically sound, and do the data support the conclusions?

Reviewer #1: Partly

Reviewer #2: Partly

2. Has the statistical analysis been performed appropriately and rigorously? 

Reviewer #1: No

Reviewer #2: Yes

3. Have the authors made all data underlying the findings in their manuscript fully available?

Reviewer #1: Yes

Reviewer #2: Yes

4. Is the manuscript presented in an intelligible fashion and written in standard English?

Reviewer #1: No

Reviewer #2: Yes

5. Review Comments to the Author

Reviewer #1: In this manuscript, 32 SARS-CoV-2 strains were isolated, of which 21 were complete genome and 11 partial. Complete genome sequences have been submitted to Genbank. Phylogenic, SNP and selection analyses have been performed for these genomes.

Comments:

(1) This manuscript was not well-organized and the English writings need to be improved, such as typos and misuse of punctuation.

(2) There should be more details about SARS-CoV-2 cases, epidemiology and evolution in Pakistan in the background part.

(3) There are 57 complete and high coverage genome sequences available from Pakistan at Sep. 31, 2000 in the GISAID EpiCoV database. These genomes need to be added for phylogenetic analyses. Moreover, other genome sequences used in analyses need description. E.g., the choose of the 449 full-length sequences, and the 6523 global complete sequences. By the way, the "supplemtary file 1" is not available.

(4) HyPHY SLAC is not adequate for selection pressure analysis. More methods need to be applied. And the gene/cds used for the analysis need to be indicated.

(5) High-resolution figures for Fig. 1 and 2 need to be provided. Phylogenetic tree should be displayed in rectangular, rather than circular.

(6) A clear conclusion need be provided. The difference and relationship of first and second wave should be discussed.

(7) Some references were not cited properly.

Reviewer #2: This manuscript presents a meaningful and useful result, and futher increases our understanding of SAR-CoV-2 in Pakistan.

However, the genomic analysis of this manuscript is not deep enough.

For example, in Fig1 and Fig2, the clades and isolate date are needed to shown in figures and analysis, the additional analysis may make the conclusion of this manuscript more credible.

Page16: Figure 3.

6. PLOS authors have the option to publish the peer review history of their article (what does this mean?). If published, this will include your full peer review and any attached files.

Reviewer #1: No

Reviewer #2: No

---

## [Author Response · Author response to Decision Letter 0]

4 Aug 2021

View Letter

Dear Dr. Ata,

Editor PLOS ONE,

Thank you for the review of our submitted manuscript:

PONE-D-21-09819

Increased diversity with absence of site-specific selection bias in SAR-CoV-2 genomes from the early pandemic period in Pakistan.

We provide responses to the questions raised by the reviewers as below:

Journal Requirements:

- thank you for sharing these links. We have revised the manuscript in accordance with the required style templates 

2.Thank you for including your ethics statement: "This work received approval by the Ethical Review Committee, Aga Khan University.". 

Please provide additional details regarding participant consent. In the ethics statement in the Methods and online submission information, please ensure that you have specified (1) whether consent was informed and (2) what type you obtained (for instance, written or verbal, and if verbal, how it was documented and witnessed). If your study included minors, state whether you obtained consent from parents or guardians. If the need for consent was waived by the ethics committee, please include this information.

 - thank you for this comment. We have described the process of testing from anonymized archived samples and Ethical Review Committee approval was received as a wavier of informed consent for the project. This has been added in the Methods section.

3.In your Data Availability statement, you have not specified where the minimal data set underlying the results described in your manuscript can be found. PLOS defines a study's minimal data set as the underlying data used to reach the conclusions drawn in the manuscript and any additional data required to replicate the reported study findings in their entirety. All PLOS journals require that the minimal data set be made fully available. For more information about our data policy, please see http://journals.plos.org/plosone/s/data-availability.

- All data is available publically. Accession numbers of the SARS-CoV-2 deposited and available are present in Methods and also in Supplementary Table 1.

- Not applicable – data is provided

- This correction has been made.

Reviewers' comments:

Reviewers Comments to the Author

Reviewer #1: In this manuscript, 32 SARS-CoV-2 strains were isolated, of which 21 were complete genome and 11 partial. Complete genome sequences have been submitted to Genbank. Phylogenic, SNP and selection analyses have been performed for these genomes.

Comments:

(1) This manuscript was not well-organized and the English writings need to be improved, such as typos and misuse of punctuation.

- We have re-organised the paper and made necessary English language corrections.

(2) There should be more details about SARS-CoV-2 cases, epidemiology and evolution in Pakistan in t-he background part.

- The background has been revised to include more details about COVID-19 in Pakistan and to also include what is know about the genomic epidemiology of SARS-CoV-2 strains from the country.

(3) There are 57 complete and high coverage genome sequences available from Pakistan at Sep. 31, 2000 in the GISAID EpiCoV database. These genomes need to be added for phylogenetic analyses. Moreover, other genome sequences used in analyses need description. E.g., the choose of the 449 full-length sequences, and the 6523 global complete sequences. By the way, the "supplemtary file 1" is not available.

- We thank the reviewer for pointing this out. We have now included all additional samples available from Pakistan on GISAID. Therefore, we have conducted analysis of 90 genomes in total.

Supplementary file 1 has been added – we apologise that it was missed last time. Corrections have been made to correctly list the number of strains which were analysed.

(4) HyPHY SLAC is not adequate for selection pressure analysis. More methods need to be applied. And the gene/cds used for the analysis need to be indicated.

 - Thank you for the advice. Additional analysis has been conducted and included in the study. These are described in methods and the results. Further, statistical analysis has been performed.

(5) High-resolution figures for Fig. 1 and 2 need to be provided. Phylogenetic tree should be displayed in rectangular, rather than circular.

- Fig 1 phylogenetic tree has been displayed as a rectangular figure. High-resolution figures have now been included.

(6) A clear conclusion need be provided. The difference and relationship of first and second wave should be discussed.

- Conclusions have been revised and the relationships between the first and second wave of COVID-19 in Pakistan have been discussed.

(7) Some references were not cited properly.

- References have been checked and cited as necessary.

Reviewer #2: This manuscript presents a meaningful and useful result, and futher increases our understanding of SAR-CoV-2 in Pakistan.

However, the genomic analysis of this manuscript is not deep enough.

For example, in Fig1 and Fig2, the clades and isolate date are needed to shown in figures and analysis, the additional analysis may make the conclusion of this manuscript more credible.

Page16: Figure 3.

- The comments of Reviewer 2 are not completely clear. We understand that the reviewer required a deeper genomic analysis. This has now been provided through the additional analysis that has been conducted.

- Further we have included additional figures to describe the clades as per the date that they were identified (S Fig1 and S Fig 2).

- Further mutational analysis has been performed regarding the entropy across the genome and also site-specific entropy. The conclusions have been justified accordingly and we hope that this will not be acceptable

---

## [Editor Report · Decision Letter 1]

9 Aug 2021

Higher entropy observed in SAR-CoV-2 genomes from the first COVID-19 wave in Pakistan

PONE-D-21-09819R1

Dear Dr. Hasan,

We’re pleased to inform you that your manuscript has been judged scientifically suitable for publication and will be formally accepted for publication once it meets all outstanding technical requirements.

Kind regards,

Sagheer Atta, Ph.D

Academic Editor

PLOS ONE
---

## [Editor Report · Acceptance letter]

17 Aug 2021

PONE-D-21-09819R1 

Higher entropy observed in SAR-CoV-2 genomes from the first COVID-19 wave in Pakistan 

Dear Dr. Hasan:

I'm pleased to inform you that your manuscript has been deemed suitable for publication in PLOS ONE. Congratulations! Your manuscript is now with our production department. 

Kind regards, 

on behalf of

Dr. Sagheer Atta 

Academic Editor

PLOS ONE